# Atlas-Based Segmentation in Extraction of Knee Joint Bone Structures from CT and MR

**DOI:** 10.3390/s22228960

**Published:** 2022-11-19

**Authors:** Piotr Zarychta

**Affiliations:** Faculty of Biomedical Engineering, Silesian University of Technology, Roosevelta 40 St., 41-800 Zabrze, Poland; piotr.zarychta@polsl.pl; Tel.: +48-32-277-74-67

**Keywords:** atlas-based segmentation, image matching, bone structures of the knee joint, femur, tibia, patella, fuzzy connectedness

## Abstract

The main goal of the approach proposed in this study, which is dedicated to the extraction of bone structures of the knee joint (femoral head, tibia, and patella), was to show a fully automated method of extracting these structures based on atlas segmentation. In order to realize the above-mentioned goal, an algorithm employed automated image-matching as the first step, followed by the normalization of clinical images and the determination of the 11-element dataset to which all scans in the series were allocated. This allowed for a delineation of the average feature vector for the teaching group in the next step, which automated and streamlined known fuzzy segmentation methods (fuzzy c-means (FCM), fuzzy connectedness (FC)). These averaged features were then transmitted to the FCM and FC methods, which were implemented for the testing group and correspondingly for each scan. In this approach, two features are important: the centroids (which become starting points for the fuzzy methods) and the surface area of the extracted bone structure (protects against over-segmentation). This proposed approach was implemented in MATLAB and tested in 61 clinical CT studies of the lower limb on the transverse plane and in 107 T1-weighted MRI studies of the knee joint on the sagittal plane. The atlas-based segmentation combined with the fuzzy methods achieved a Dice index of 85.52–89.48% for the bone structures of the knee joint.

## 1. Introduction

The knee joint is a large and complex human joint. It is formed by the following bone structures: distal end of the femur, proximal surface of the tibia bone, and the patella. The distal end of the femur (from a medical point of view) rests and slides on the proximal surface of the tibia bone, whilst the patella (a flat bone located in front of the knee) slides on the front surface of the distal end of the femur. The whole knee joint is completed by ligaments and muscles [1,2].

Two convex condyles of the femur form the joint head. The concave surfaces of the tibial condyles together with the patella form the acetabulum of the knee joint. In the case of the joint head, the joint surfaces of both condyles are strongly convex in the sagittal axis and slightly in the frontal axis. The front parts appear flattened in relation to the strongly rounded posterior portions. It is on these posterior parts that the rotational movement of the tibia occurs when the knee is bent. The joint surfaces of the medial and lateral femoral condyles are connected by the patellar surface. In the case of the acetabulum of the knee joint, the joint surface of the tibial bone medial condyle is oval as well as larger and deeper than the joint surface of the lateral condyle. The joint surface of the lateral condyle is triangular. The surface of the acetabulum formed by the tibial condyles is about three times smaller than the femoral joint head. It is supplemented by the articular surface of the patella [1].

The proper bone segmentation of the knee joint is an important issue in the new approach to knee replacement [3] and also in the case of patellar chondromalacia [4,5,6].

This new proposed approach to knee replacement is firstly based on the automatic determination of the mechanical axis of the lower limb, and then also on the automatic determination of the cutting planes of both the femur and the tibia. The automatic determination of the mechanical axis is realized before surgery based on the toposcan of this limb. The cutting planes are indicated based on the imprints of both bone heads with marked Kirchner holes, which is also performed before surgery. The required imprints are obtained on the basis of bone structures extracted from the CT (Computed Tomography) or MRI (Magnetic Resonance Imaging) slices of the knee joint. The segmentation of the proper bone structures is a fundamental element of this approach. The proposed approach to knee replacement surgery positively affects two essential risk factors in the following ways: it reduces exposure to infections (by shortening the amount of time needed for surgery by about 50%; the mechanical axis is determined automatically, not manually), and more importantly, it eliminates disturbances of the mechanical axis of the lower limb. From the practical (medical) point of view, this new approach to knee joint arthroplasty may be particularly helpful for an inexperienced orthopedist, such as in cases of doubt concerning the determination of the mechanical axis of the lower limb. A small error (even of a few degrees) in the process of determining the mechanical axis may result in the erroneous fixation of endoprosthesis implants.

Under physiological conditions, the patella moves in the groove of the knee joint. At the moment when the patella escapes from the groove of the knee joint, pain comes and problems with walking appear. The patella significantly affects the proper functioning of the entire knee joint. In medical practice, orthopedists usually encounter several specific problems related to the patella:Excessive lateral pressure syndrome (ELPS);Dysfunctions of the patella retinaculum;Dysfunctions of the patellofemoral joint;Subluxation of patella;Recurrent dislocation of patella;Patellar chondromalacia.

The most frequent of these diseases include patellar chondromalacia and excessive lateral pressure syndrome. Each diagnosis should be based on functional diagnostics performed by an experienced orthopedist. In the resolution of the cause of the problem, imaging studies are indispensable (Radiography, UltraSonoGraphy, CT, MRI). An MRI study of the patellofemoral joint is more accurate due to the fact that it also shows the actual joint surfaces (cartilage) and not only the bony ones, as in radiography. An important advantage of MRI is the ability to show cartilage defects. In the case of the patella, the most frequent problem is the occurrence of stiffness and pain in the front part of the knee during long sitting, running, walking on stairs, squatting, or kneeling. Patients often talk about such symptoms as "the patella jumping in the knee". The pain often seems to be spilled as well. Therefore, it is difficult for the patient to identify the specific place of its occurrence. It is very important, however, not to underestimate such symptoms. Unfortunately, disregarding such symptoms for a long time increases the risk of dislocation of the patella; it can also lead to the damage of the patellofemoral cartilage, and thus to the chondromalation of the patella.

From a medical point of view, patellar chondromalacia is a disease of the cartilage that surrounds it [1,4,5]. The wrong angle of movement of the patella causes the patella to rub against the bone instead of sliding. In this case, it is usually very helpful for an orthopedist (especially in doubtful instances) if the patellar structures can be extracted from MRI slices or CT scans and if its 3D presentation can be obtained. A 3D presentation on any plane allows the medical doctor to accurately diagnose the patella.

In the literature, many different methods dedicated to the segmentation of various anatomical structures are described. In most cases, these are the user interaction methods. From the medical point of view, automatic methods are very desirable. With respect to the knee joint, an elementary and first step is the accurate segmentation of the bone surface. This will allow for a reliable segmentation of the remaining anatomical structures of the joint [2,7] or other medical procedures associated with this joint [3]. In practical solutions, the following methods are used [8]:Thresholding;Region growing;Clustering;Deformable models;Atlas;Deep learning.

Methods based on thresholding are intensity-based. These methods are not time-consuming and can be applied globally or locally. Usually, on the basis of single or multiple thresholds, tissues of the knee joint can be extracted in an approximate way [8].

Methods based on region growing are region-based segmentations, which are locally applied on the image. These approaches examine pixels in the neighborhood of the start points (seed points) and merge them if a homogeneity condition is fulfilled [8].

Clustering methods are usually used for segmentation based on the MRI slices or CT scans of the human body. The clustering approaches divide the feature space of an image into clusters, and the clusters correspond to data that have a certain level of similarity. Kmeans and Fuzzy CMeans [2] are quite widely used algorithms in these methods.

Methods based on deformable models usually require user interaction and are semi-automated. However, these methods are widely used in clinical applications. It should be noted that the segmentation is obtained by deforming an elastic contour, and this contour evolves toward the searched object driven by image forces. The following algorithms are quite widely used in these methods: snakes and active contours [8,9].

According to the literature [8,9], atlas-based segmentation methods provide results at levels above 85–95% of the Dice index (for the femoral cartilage, 63.7–88% [10,11,12,13,14]; for the tibial cartilage, 65.5–84% [10,11,12,13,14]; for the tibia, 84–96% [10,13,14]; for the femur, 88–96.9% [10,13,14]). These are satisfactory values, although they require a lot of work from the expert (radiologist). The preliminary research works carried out gave grounds to suppose that the use of this method in order to determine, for example, the starting points (seed points) or the initial region of interest (containing cruciate ligaments) and then transfer the information to implement the fuzzy methods would yield promising results.

In the course of the last years, approaches based on Machine Learning and Deep Learning have attracted interest. Currently dominating in practical applications is Deep Learning (revealing high accuracy and fast computational time compared to state-of-the-art methods), with a model of the convolutional neural network (CNN) and its architectures in several applications. With regard to the structures of the knee joint, Deep Learning methods have been applied to the following anatomical structures of this joint:Bone structures: In [15], a fully automatic model was proposed to detect and segment knee bones using modified U-net models. The obtained accuracy of the bone structures was equal to 98%, and the Dice index for the patella: 92%, tibia: 96%, and femur: 97%. In [16], a 2D convolutional encoder network of a Visual Geometry Group 16 (VGG16) architecture (Dice index for femur: 96% and for tibia: 95%) was used.Anterior cruciate ligament: In [17], a customized, 14-layer residual network ResNet-14 architecture of the CNN was used with six different directions by employing class balancing and data augmentation. The obtained accuracy was equal to 92%. In [18], a self-supervised approach was proposed, with pretext and downstream tasks using class balancing through oversampling (accuracy was equal to 90.6%). In [19], the AlexNet architecture of the CNN to extract features of the knee MRNet with transfer learning ImageNet was presented (accuracy was equal to 93.7%). In [20], a knee mask was generated on the original MRI images to apply a semantic segmentation technique with the CNN architecture U-Net (accuracy was equal to 98%, and the Dice index was 99%).Cartilage of the knee: In [21], 2D features of the CNN were used for each voxel of three planes (accuracy: 99.9% and Dice index: 82%). In [16], a 2D convolutional encoder network of a Visual Geometry Group 16 (VGG16) architecture was used (Dice index for femoral cartilage: 81%, and for tibial cartilage: 82%).

Deep Learning methods, apart from their undoubted advantages, also have disadvantages. The right structure of the network has to be chosen. A network that is too small, with no hidden layers, loses its ability to solve problems; even a long training period is not able to help. Too many hidden layers lead to a simplification of the network performance. Even minor errors in labeling the training data, which can occur quite frequently due to human error, can ruin the accuracy of the neural network. In addition, huge amounts of data are needed for training. The training process itself is computationally very expensive, requires a large amount of memory and computing resources, and transferring it to other problems is not easy. As a result, the algorithm takes longer to train, and more memory is required to work with the data.

Therefore, considering the information above, the novelty of this document is in the automatic extraction of bone structures of the knee joint—the femoral head, tibia, and patella—based on atlas segmentation and the use of an automated image-matching methodology (an approach using the fuzzy image concept instead of the standard four main steps of the registration process: feature detection, feature matching, mapping function design, and image transformation with resampling). This method allows for the construction of a feature vector that automates and streamlines known segmentation methods (e.g., fuzzy c-means (FCM), fuzzy connectedness (FC)). This paper is a continuation of pilot studies shown in [22,23].

The paper is organized as follows. Section 2 describes the detection of the average feature vector (based on atlas segmentation and medical image-matching) of the femoral head, tibia, and patella. In Section 3, an evaluation of the results is presented, while Section 4 discusses the results and Section 5 concludes the paper.

## 2. Materials and Methods

### 2.1. Materials

The proposed methodology was tested on 107 clinical T1-weighted MRI studies of the knee joint (17–24 scans on the sagittal plane per volume) and 61 clinical CT studies of the lower limb (61 toposcans and 61 series of CT images, 25–33 scans on the transverse plane per volume). The MRI data were acquired on the transverse and sagittal plane for 66 females and 41 males at different ages. The CT data were acquired for 39 females and 22 males at different ages.

### 2.2. Workflow

The workflow (Figure 1) starts with an automated medical image-matching, which is followed by the normalization of a series of clinical images of the knee joint. Then, the 11-element dataset is determined, to which all scans in the series are allocated. This allows for a delineation of the average feature vector for the teaching group. From a practical point of view, two features are important. These are the centroid and the surface area of the segmented bone structure. These averaged features are subsequently transmitted to the FCM or FC methods (fuzzy methods) implemented for the testing group and respectively for each scan. The centroids then become the starting points (seed points) of the fuzzy methods, while the surface area protects the methods against over-segmentation. The problem of over-segmentation is very important, especially in the case of patella segmentation.

### 2.3. Average Feature Vector

The following steps were accomplished in order to find the average feature vector for the teaching group:Medical images matching (Figure 1b);Extraction of bone structures by an expert;Creation of feature vector for each scan in the volume;Normalization of scans within the volume (Figure 1c);Division of designated features into 11 sets (Figure 1c);Averaging of features within each of 11 sets (Figure 1d).

The application of an automated image-matching methodology [23] is based on the fuzzy image concept (entropy measure of fuzziness), which is combined with the use of similarity measures (intensity-based measures) instead of the four elementary steps of the registration process (feature detection, feature matching, mapping function design, and image transformation with resampling). In short, the highlighting of the parts of the medical image that have a significant difference in the intensity of the neighboring pixels may be expressed as entropy (or energy) measure of fuzziness (Figure 2). In this paper, the methodology of finding the entropy (or energy) measure of fuzziness is not described in detail. An exhaustive description of this method can be found in [3,24]. An assessment of the similarity of the slices in whole series is the second step in the image-matching methodology. The literature review shows that there are two main groups of similarity measures: one based on the features and the other on intensities [23,24]. Feature-based measures may be defined as the measures using the information (extracted features) about the objects obtained in the image processing. Intensity-based measures may be defined as the measures using the information determined on the basis of gray levels in both images. The advantage of intensity-based measures is the option to skip the usually time-consuming and complicated phase of feature extraction. For this reason, in this research work, the following intensity-based measures have been taken into consideration: normalized cross-correlation (NCC), gradient correlation (GC), and gradient difference (GD). In this paper, the similarity measures (intensity-based measures and feature-based measures) are not described in detail. An exhaustive description of these measures can be found in [23,24]. In short, the matching process of two different scans on the same plane and modality involves the fuzzyfication process of the original scans and then the calculation of the similarity measure between those scans. The highest value of the similarity measure indicates the most similar scans.

The extraction of bone structures (especially in the case of the portion of scans with the patella) by an expert, as a rule, involves a lot of work (many series of slices or scans have to be analyzed). This process is time-consuming but nonetheless necessary in order to build a large elementary atlas.

In the following step, the creation of a feature vector for each scan in the volume is performed. In this work, the following features are calculated: centroid, skeleton, edges, area and perimeter of the extracted bone structures as well as the area of the entire knee.

The next step in this approach is the scan normalization within the volume. This step is important due to the different number of slices or scans in each volume. The standard MRI study consists of 17–24 slices on the sagittal plane per volume, and about 25–33 scans on the transverse plane per volume for the CT study.

After the scan normalization, the division of designated features into 11 sets is carried out in the next step. This process is implemented in order to adapt the determined features to their faithful representation in the atlas-based segmentation. The division into 11 sets is the result of simulations that take into account the layer thickness in the MRI and CT studies; above all, this results from a compromise between the acceptable effectiveness of the segmentation based on the fuzzy methods and the time required to determine the features within the individual sets (Figure 3).

The last step allows for the average features to be obtained by each of the 11 sets.

Figure 4 shows the operation of superposition for the selected 3rd, 6th, and 9th set of the teaching group on the transverse plane together with the designated centroid for the patella, femur, and tibia, respectively.

Quite a similar situation is presented for the sagittal plane (Figure 5). This figure presents the operation of superposition for the selected 3rd, 6th, and 9th set of the teaching group on the sagittal plane together with the designated centroid for the patella, femur, and tibia, respectively.

### 2.4. Fuzzy Segmentation Methods

The extraction process of the selected anatomical structures of the human body is usually one of the stages in computer-aided medical diagnostics. According to the literature, this process can be carried out as follows: manually (usually by a medical expert), interactively (usually performed in cooperation with a medical expert), or fully automatic. From a medical point of view, the interactive or fully automatic realization of the process is preferred. The manual extraction process of the selected anatomical structures is very complex, usually tedious and time-consuming, and it may sometimes be subject to relatively large errors. In the literature, over the last ten years, two dominant types of methods for the extraction of anatomical structures of the human body can be found. These extraction methods are those dedicated to selected organs of the human body and general methods. After the selection of suitable method parameters, the general methods can usually be employed to extract any anatomical structure. In this work, in order to automatically extract the bone structures of the knee joint, the following fuzzy segmentation methods have been taken into consideration: FCM and FC. In both of these methods, the sets of centroids (corresponding to a given scan) obtained in the course of the atlas-based segmentation for the patella, femur, and tibia were used as starting points (Figure 4).

The fuzzy methods mentioned above were chosen due to the fact that fuzzy logic is quite a good tool for the computer-aided diagnosis of the knee joint. In fact, the use of classical methods is usually effective in the case of artificial image analysis. Unfortunately, these methods do not give correct results in the analysis of medical images. Generally, the medical images are characterized by their very large complexity. This complexity requires taking into consideration many elements that usually influence each other. In this case, two important problems arise: inaccuracy and data uncertainty.

In this paper, the FCM and FC segmentation algorithms are not described in detail. A detailed description of these algorithms can be found in [3].

## 3. Results

### Experiments and Evaluation

The proposed automated image-matching process based on the fuzzy image concept, which was combined with the use of similarity measures in order to find the most similar MR scans in the inter-object matching (scans containing bone structures of the knee joint), was tested on 107 clinical T1-weighted MR studies of the knee joint on the sagittal plane. The obtained results were verified by two independent experts, and correct results (for the inter-object matching of the whole scans containing the bone structures of the knee joint) were yielded in 84 cases (the accuracy of the method is equal to 78.5%).

Figure 6 shows the results of the matching process for two selected MR scans containing the bone structures of the knee joint. In both cases (scan no.5 and scan no.8 of the first group), the highest value of the similarity measure indicates the most similar scan (scan no.6 and scan no.10) from the second group.

In the next step, the obtained dataset of clinical scans was divided into two groups. These were the teaching and the testing groups. The teaching group consisted of 50 arbitrarily selected clinical T1-weighted MRI studies of the knee joint and 30 clinical CT studies of the lower limb. Based on this group, a set of centroids, edges, areas, and perimeters from the extracted bone structures (corresponding to a given scan) for the patella, femur, and tibia as well as a set of areas of the entire knee joint (with respect to a given scan) were determined. The rest of the obtained clinical dataset constituted the testing group.

Based on the images of the teaching group (50 clinical T1-weighted MRI studies of the knee joint and 30 clinical CT studies of the lower limb), the normalized values of the x and y coordinates of the centroid (patella, femur, and tibia) were obtained in view of the size of the scans on the transverse plane (Table 1) and on the sagittal plane (Table 2). In the next step, the x and y coordinates were used as starting points (seed points) in both implemented methods (FCM and FC).

In the case of the transverse plane, the determined centroids proved to be excellent starting points for the bone structures of the knee joint. The situation was much worse in the case of the sagittal plane, especially for the bone structures of the patella (Figure 5). From the practical point of view, the centroids of the patellar structures determined on the basis of the teaching group for 11 sets had to be modified before using them as starting points. This modification consisted in choosing such a point that belonged to the skeleton as the starting point, for which the distance between the average centroid and this point was the smallest. The choosing operation was carried out until the area of the obtained patellar extraction conformed with the area given in (Table 3) for each set. In the case of the femur and tibia on the sagittal plane, such a modification was not necessary.

In Table 3 and Table 4, the normalized (to the size of the knee) areas of the patella, femur, and tibia structures on the transverse and sagittal planes, respectively, are given. These areas were used in the extraction of the bone structures from the T1-weighted MRI studies of the knee joint and the CT studies of the lower limb as a protection against over-segmentation and in the process of shifting the designated centroids of the patella on the sagittal plane (Table 4).

The final stage in the automatic extraction of bone structures of the knee joint from MRI and CT scans based on the atlas segmentation was accomplished by means of post-processing. The post-processing operation consisted in performing consecutive steps, the aim of which was to obtain separate patellar, femoral, and tibial structures with smoothed edges. In order to achieve this, the operation was based on averaging filtration combined with the implementation of morphological operations. This approach, in the most typical cases, resulted in the correct extraction of bone structures of the knee joint.

Atlas-based segmentation combined with the fuzzy connectedness method applied to bone structures of the knee joint gave the following Dice index results (average values for the testing group): 89.48% for the patella, 86.59% for the femur, and 87.94% for the tibia, with regard to the clinical CT studies of the lower limb. For atlas-based segmentation combined with the FCM method, similar values were obtained. For the same clinical CT studies, atlas-based segmentation combined with the FCM method applied to the same anatomical structures of the knee joint gave Dice index results (average value for the testing group) at the following levels: 88.23% for the patella, 85.75% for the femur, and 87.14% for the tibia.

Atlas-based segmentation combined with the same fuzzy methods (FC and FCM), but applied to bone structures for T1-weighted MRI studies of the knee joint on the sagittal plane, gave the following Dice index results (average value for the testing group): 87.59% (FC) and 86.65% (FCM) for the patella, 88.66% (FC) and 87.49% (FCM) for the femur, and 86.59% (FC) and 85.52% (FCM) for the tibia.

The obtained Dice index values for the analyzed CT scans (testing group) are presented in Table 5 (atlas-based segmentation combined with the FC and FCM methods), and those for the MRI scans are shown in Table 6.

The discrepancy in the obtained values of the Dice index between the atlas-based segmentation combined with the FC method and the atlas-based segmentation of the same bone structure combined with the FCM method for CT clinical data is illustrated by means of a box-and-whisker plot (Figure 7) and Bland–Altman plots (Figure 8). Further on, the box-and-whisker plot (Figure 9) and the Bland–Altman plots (Figure 10) show the discrepancy of the obtained Dice index values between the atlas-based segmentation in combination with the FC method and then with the FCM method; both methods were tested here for clinical T1-weighted MRI studies of the knee joint.

After the tests described above, the phantom tests were carried out. In these studies, atlas-based segmentation was used, leading to the creation of femoral imprints [11]. Firstly, a CT study of the artificial femur was performed. Then, atlas-based segmentation combined with the FC method was used in order to perform femoral head extraction, after which a 3D model of this structure was created (Figure 11c). Finally, an imprint of this artificial structure was made. The imprint thus obtained was applied to the artificial femur, and two independent experts (radiologist and orthopedist) assessed the degree of fit. In their opinion, the degree of fit achieved by the imprint was sufficient. There was no slack, rotation, nor sideslip, and there was no lateral movement either. The central and side surfaces of the artificial femur closely adhered to the imprint. The artificial femur was stable and remained motionless.

The following tests were performed in the described studies: the Wilcoxon test and the *t*-test. The reason for this step was the lack of distribution normality for the compared variables. The obtained results from the performed statistical tests clearly show the small difference between the results obtained from the two implemented fuzzy approaches, FC and FCM. The analysis of the calculated *p*-values for the Wilcoxon test and the *t*-test (Table 7) led to the following conclusions: The difference between both methods (FC and FCM) is not statistically significant (for all cases, the calculated *p*-values were lower than 0.05). The described method has acceptable performance.

Table 8 shows the following: mean, standard deviation, and *p*-values, calculated by using the Bland–Altman plots, followed by a one-sided *t*-test for the patella, femur, and tibia. Furthermore, in this situation, for all cases, the calculated p-values were much lower than 0.05, indicating that the difference between the FC and FCM measurements is not statistically significant, and that both proposed fuzzy methods performed quite well.

The data (10 MRI T1-weighted series on the sagittal plane) used to verify the results of this study were obtained from the NYU fastMRI Initiative database (fastmri.med.nyu.edu) [25]. As starting points for both fuzzy methods (FC and FCM), those determined in this study (Table 2) were used. The obtained values of the Dice index for the analyzed MRI series are presented in Table 9. It can be noted that the atlas-based segmentation combined with the same fuzzy methods (FC and FCM) but applied to bone structures for the fastMRI Dataset (10 T1-weighted MRI studies of the knee joint on the sagittal plane) gave the following Dice index results (average value for the testing group): 86.18% (FC) and 85.33% (FCM) for the patella, 88.19% (FC) and 87.55% (FCM) for the femur, and 87.12% (FC) and 86.57% (FCM) for the tibia. These values do not differ significantly from those obtained in the study, therefore the results can be considered as convergent and reliable.

## 4. Discussion

On the basis of testing studies performed on 107 clinical T1-weighted MRI studies of the knee joint (on the transverse and sagittal planes) and 61 clinical CT studies of the lower limb, the following elements can be proven. The described automated methodology dedicated to the extraction of bone structures from MRI and CT scans based on the atlas segmentation seems to be very effective and promising.

From the group of direct similarity measures, normalized cross-correlation was used, primarily because the contrast and brightness values of the compared images did not negatively affect the quality of the similarity measures (except for differences caused by rounding off the value of intensity). The sum of squared difference (SSD) and the sum of absolute difference (SAD), which also belong to the group of direct similarity measures based on intensity, were omitted from the study due to their basic defect—the possibility of falsifying the results in the case of even a small number of pixels showing significantly different levels of intensity.

From the group of similarity measures based on spatial information, the following measures were chosen: gradient correlation (GC) and gradient difference (GD). These measures were selected on the basis of their undeniable advantage—the ability to perform a correct evaluation of the compared images with both soft structures (joints) and hard structures (bones, intervention tools). The scaling factor for the gradient difference had to be determined so that the difference image could be characterized by the lowest contrast (the analyzed images had an identical size of 256 × 256, so the scaling factor s = 1). In the computer simulations performed, the horizontal and vertical Sobel masks of size 3 × 3 were used. These were linked with the length of the calculation time since increasing the size of the mask (from 3 × 3 to 7 × 7) brought about a slight improvement in the results, but it was associated with a significant lengthening of the duration of calculation.

The sum of the local normalized correlation (SLNC) and pattern intensity (PI), which also belong to the group of similarity measures based on spatial information, was omitted in this work, mainly due to the factor related to the time needed to perform the necessary calculations.

A disadvantage of the mutual information (MI), which resulted in its exclusion from the registration of two different series of MRI or CT, was the large time-consumer of the optimization process (the optimization function has a number of local optima and a single significant global optimum). The second main disadvantage of the similarity measures based on the histogram is the assumption that the degree of blurring of the histogram depends on the degree of mismatch of the images, which is why no account of spatial information was taken.

In the proposed approach, after the operation of scan normalization, the division of designated features into 11 sets was subsequently carried out. Such a division allowed for the acquisition of a DICE value of almost 90 percent. For a larger number of sets, the increase in this value is relatively insignificant, but the calculation process entails a significant increase in the time required to determine the features within individual sets. Therefore, from the point of view of adapting the designated features to their most faithful representation in the process of atlas segmentation, the 11-element set is acceptable. It takes into account two aspects: the layer thickness in the MRI and CT examinations and a compromise between the acceptable effectiveness of the segmentation method and the time required to determine the features within individual sets.

The obtained centroids of particular bone structures of the knee joint, determined on the basis of the atlas method, proved to be very effective as starting points. In the case of the transverse plane, a small problem appeared during the femur segmentation for set 11 (the last scans in the volume containing the deepest intercondylar fossa). Over-segmentation often occurs here, especially for persons with a degenerative disease and indications for partial or total alloplasty [1,3]. The use of normalized information for two centroids $x_{f1}$, $y_{f1}$ and $x_{f2}$, $y_{f2}$ (Table 1), and additionally for the surface area (Table 3), leads, in the most typical case, to the elimination of this inconvenience.

A similar situation occurs in the case of the tibia bone on the same plane. On the upper surface of the proximal end, the tibia has two concave upper joint surfaces. These concave upper joint surfaces lie on the medial condyle and lateral condyle. Between them, there is the intercondylar tibial eminence. An anterior intercondylar field is located forward of the tibial eminence, while a posterior intercondylar field is located to the rear of the tibial eminence [1,3]. For this reason, in the case of tibia segmentation for set no.1, over-segmentation often occurs. The use of normalized information for two centroids $x_{t1}$, $y_{t1}$ and $x_{t2}$, $y_{t2}$ (Table 1), and additionally for the surface area (Table 3), leads, in the most typical case, to the elimination of this inconvenience.

According to the literature [26,27], in the field of clinical medicine, the measurements made on a living human body undergo constant changes. The true value of these measurements very often remains unknown. This implies the necessity for continuous improvement and the creation of new and better measuring methods, tools, and applications. From the practical point of view, the best way to proceed is to compare the compatibility of a new method with the old method (or the method used so far). Usually, in the field of medicine (i.e., biomedical images), one cannot expect that the new method will give exactly the same result as the method used so far. It is important, however, to check how different the results are. In order to replace the old method with a new one, the obtained difference between the results of both methods should be small enough not to create a problem in the clinical interpretation. Statistical methods will not answer the question as to how large the difference in the methods could be for the methods to be considered compatible. Consequently, an appropriate graphic illustration of the differences obtained and the possible limits of variability is required, as shown in Figure 7, Figure 8, Figure 9 and Figure 10. On the grounds of the above figures, it can be concluded that in the case of bone structures of the knee joint, the atlas-based segmentation in combination with the FC method gives slightly better results (about 1% Dice index) than the same atlas-based segmentation in combination with the FCM method. Of course, the above thesis concerns the research group (materials) of CT and MRI images.

## 5. Conclusions

In conclusion, this new approach for the automatic extraction of bone structures of the knee joint: femoral head, tibia, and patella, which is based on the atlas method, seems to be quite promising. In fact, the atlas-based segmentation requires a lot of work from the expert; however, in this case, the Dice index results obtained for the MRI scans are at 85.52–88.66% and at 85.75–89.48% for the CT scans. In the medical field, these are satisfactory values.

At this point, two important problems related to the segmentation of anatomical structures are worth mentioning. The first one relates to the lack of possibility for a precise description of the segmented structures (inaccuracy). The second one concerns identical results, which are difficult to obtain (data uncertainty). The knee joint is relatively easy to identify on the image. However, it is difficult for an expert to create two identical outlines of the same bone structures of the knee joint, especially on slices containing the patella [2]. Therefore, the Dice index value at the level of 0.9 achieved in this work is not low.

Based on the obtained results, it can be concluded that the fuzzy methods, FC and FCM, can be used in the segmentation of the bone structures of the knee joint. Atlas-based segmentation combined with the FC method applied to bone structures of the knee joint gave slightly better results (Dice index of about 1%) compared to the FCM method. The results obtained using both methods are acceptable and satisfactory in medical applications.

The studies described above are being continued, and on the basis of the presented methodology, a software application has been built, which constitutes a successive step in computer-aided diagnostics of the knee joint [2,22,24,28] and in the diagnosis of pathologies of the lower limb, especially in the alloarthroplasty of the knee joint [3]. It is quite probable that in the near future, this approach may become a very useful tool in the hands of the orthopedist.

## Figures and Tables

**Figure 1 sensors-22-08960-f001:**
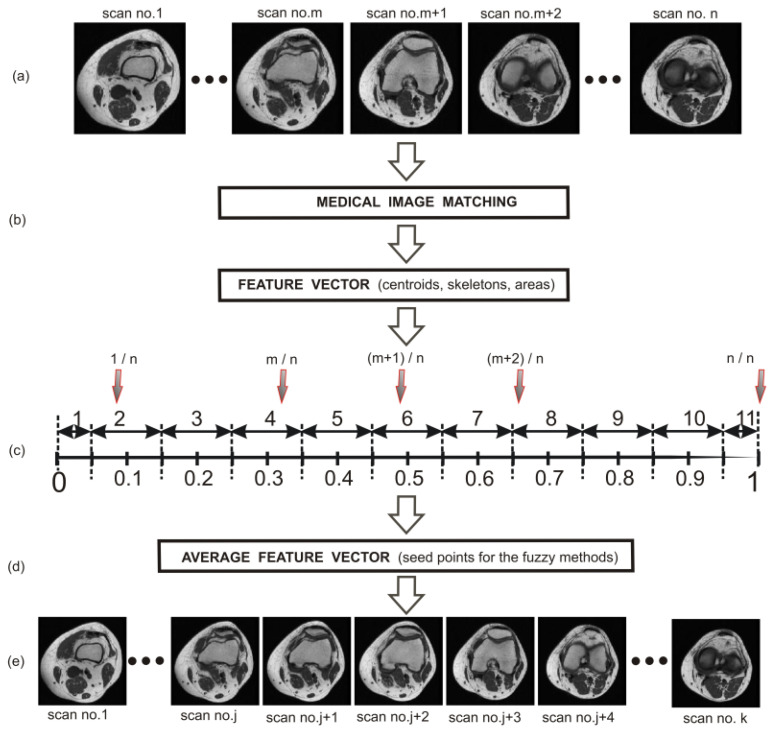
Block diagram: (**a**) Teaching group, (**b**) Automated medical image-matching, (**c**) Determining diagram of the average feature vector, (**d**) Average feature vector, and (**e**) Testing group.

**Figure 2 sensors-22-08960-f002:**
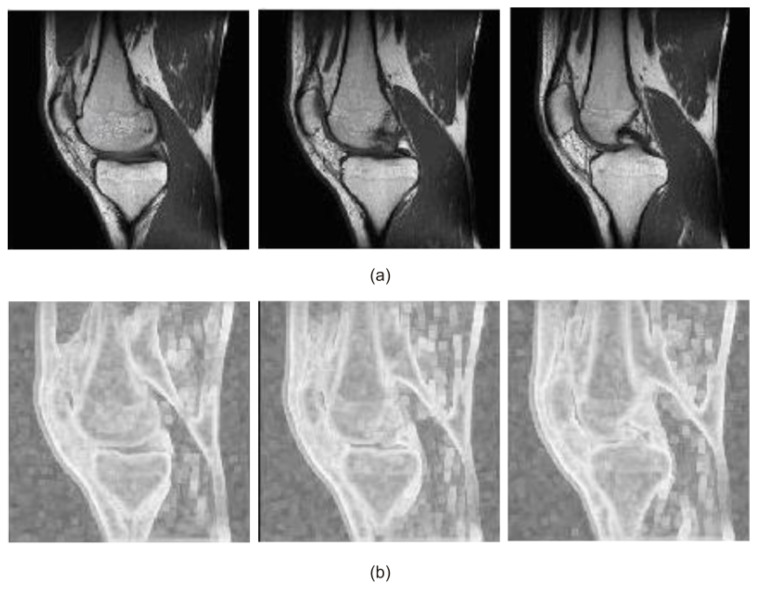
Clinical T1-weighted MRI studies of the knee joint (selected scans on the sagittal plane): (**a**) original signal and (**b**) entropy measure of fuzziness.

**Figure 3 sensors-22-08960-f003:**
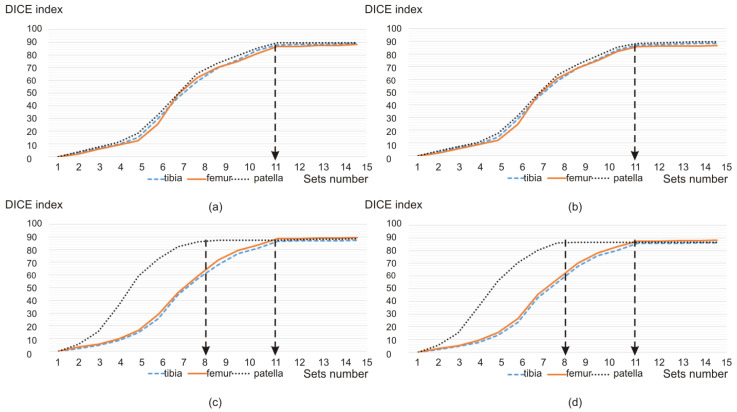
Dependence of the DICE index value on the sets number for (**a**) CT scans on the transverse plane and the FC method, (**b**) CT scans on the transverse plane and the FCM method, (**c**) MRI scans on the sagittal plane and the FCM method, and (**d**) MRI scans on the sagittal plane and the FCM method.

**Figure 4 sensors-22-08960-f004:**
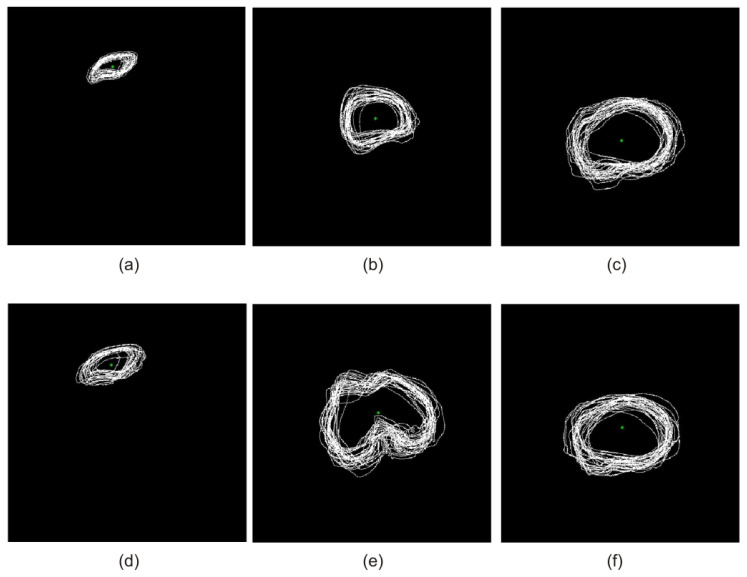
Superposition of selected 3rd (**a**–**c**), 6th (**d**–**f**), and 9th (**g**–**i**) set of the teaching group on the transverse plane along with the centroid marked for the patella, femur, and tibia, respectively.

**Figure 5 sensors-22-08960-f005:**
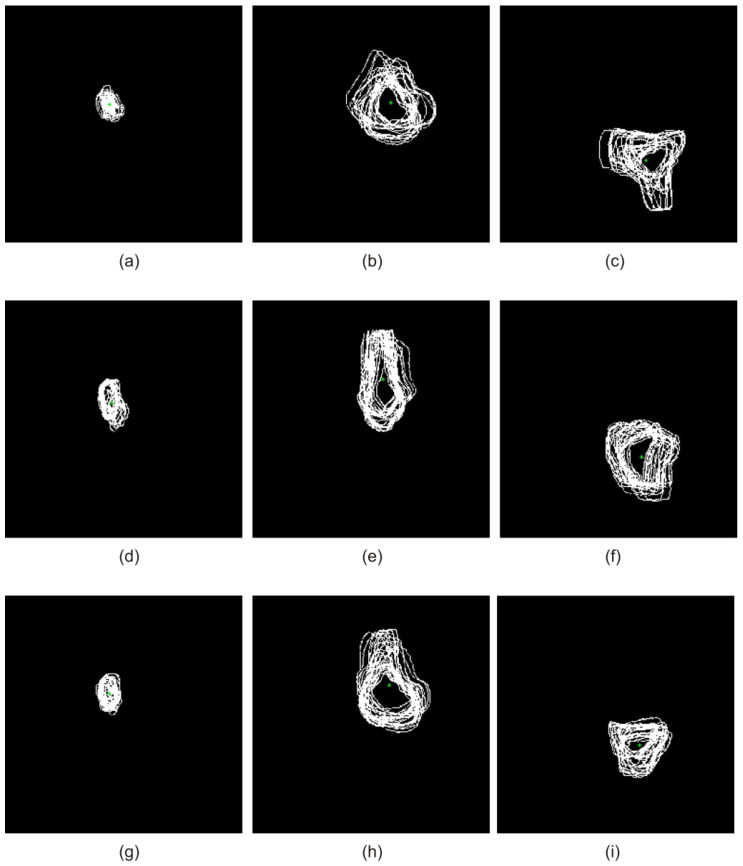
Superposition of selected 3rd (**a**–**c**), 6th (**d**–**f**), and 9th (**g**–**i**) set of the teaching group on the sagittal plane along with the centroid marked for the patella, femur, and tibia, respectively.

**Figure 6 sensors-22-08960-f006:**
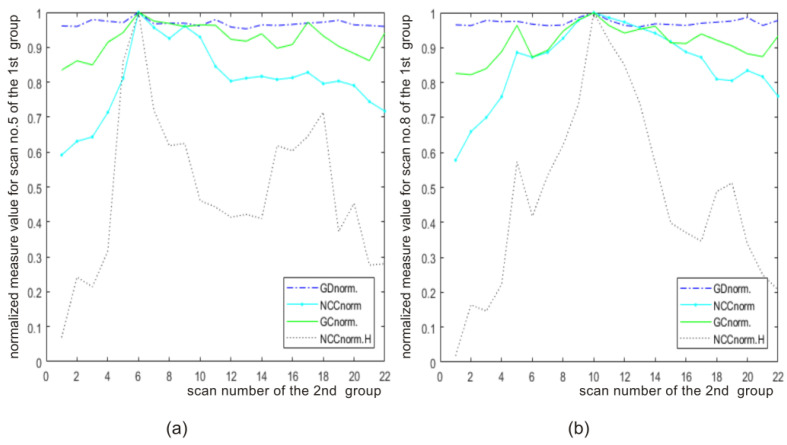
Similarity measures for scan (**a**) no.5 and (**b**) no.8 of the reference series (scans containing the bone structures of the knee joint (1st group) and tested series (2nd group).

**Figure 7 sensors-22-08960-f007:**
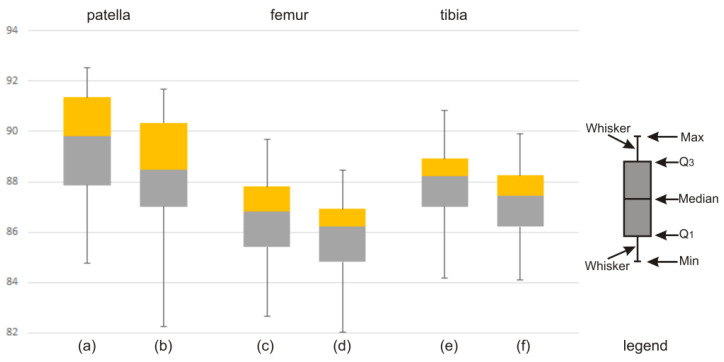
Discrepancy in the obtained values of the Dice index: (**a**,**c**,**e**) atlas-based segmentation combined with FC method (CT studies of the lower limb) and (**b**,**d**,**f**) atlas-based segmentation combined with FCM method (CT studies of the lower limb) for patella, femur, and tibia, respectively.

**Figure 8 sensors-22-08960-f008:**
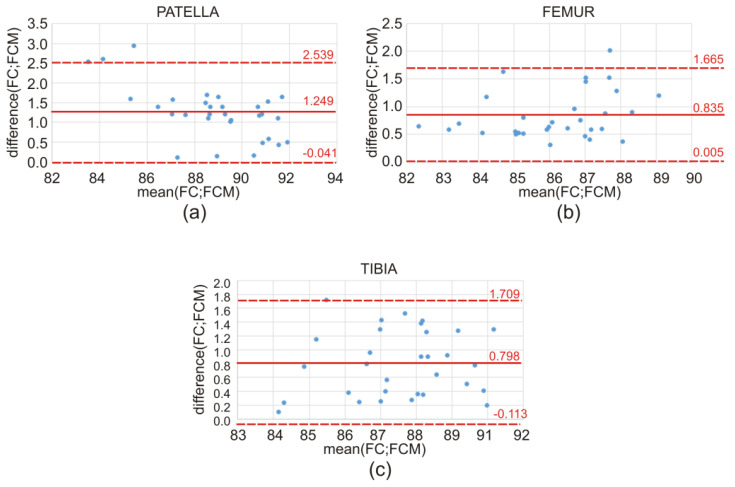
Discrepancy in the obtained values of the Dice index between the FC and FCM methods for CT studies of the lower limb on the transverse plane, illustrated using Bland–Altman plots, for the (**a**) patella, (**b**) femur, and (**c**) tibia, respectively.

**Figure 9 sensors-22-08960-f009:**
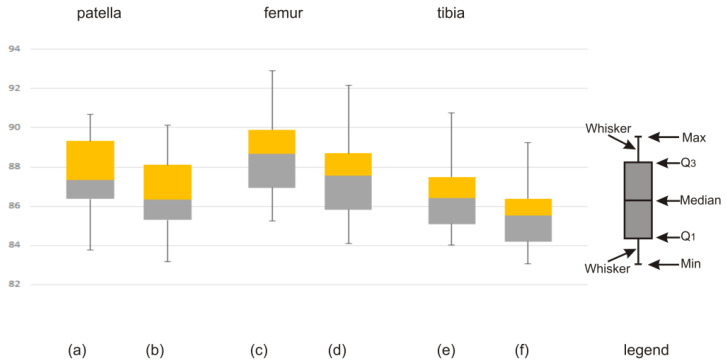
Discrepancy in the obtained values of the Dice index: (**a**,**c**,**e**) atlas-based segmentation combined with FC method (T1-weighted MRI studies of the knee joint) and (**b**,**d**,**f**) atlas-based segmentation combined with FCM method (T1-weighted MRI studies of the knee joint) for patella, femur, and tibia, respectively.

**Figure 10 sensors-22-08960-f010:**
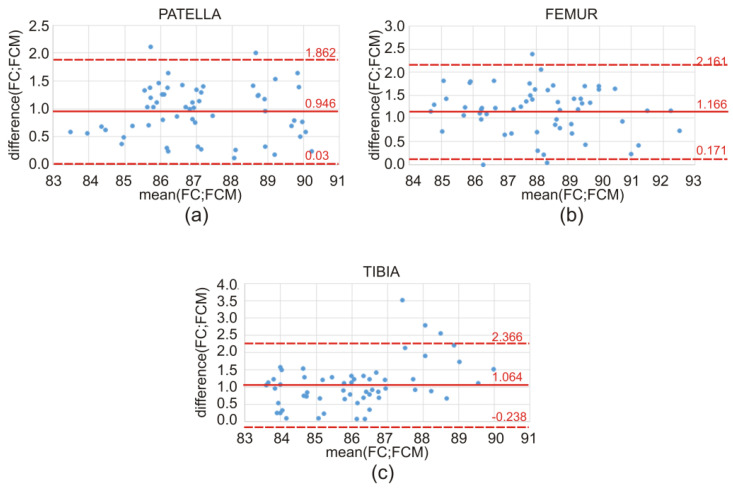
Discrepancy in the obtained values of the Dice index between the FC and FCM methods for T1-weighted MRI studies of the knee joint on the sagittal plane, illustrated using Bland–Altman plots, for the (**a**) patella, (**b**) femur, and (**c**) tibia, respectively.

**Figure 11 sensors-22-08960-f011:**
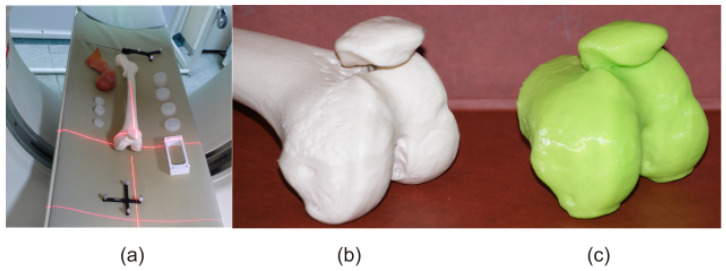
Artificial femur: (**a**) CT study, (**b**) femoral head, and (**c**) 3D model of femoral head.

**Table 1 sensors-22-08960-t001:** Normalized values of the x and y coordinates of the centroid due to the size of the slice on the transverse plane.

Set	Patella		Femur		Tibia	
	x	y	x	y	x	y
1	0.398	0.252	0.512	0.473	x ${}_{t1}$ = 0.635	y ${}_{t1}$ = 0.477
					x ${}_{t2}$ = 0.522	y ${}_{t2}$ = 0.574
2	0.428	0.250	0.514	0.470	0.505	0.578
3	0.443	0.254	0.517	0.465	0.505	0.558
4	0.436	0.256	0.521	0.471	0.508	0.553
5	0.436	0.258	0.529	0.484	0.509	0.543
6	0.453	0.271	0.522	0.487	0.521	0.543
7	0.478	0.285	0.518	0.490	0.552	0.533
8	0.453	0.277	0.518	0.478	0.514	0.555
9	0.441	0.276	0.519	0.451	0.517	0.584
10	0.436	0.279	0.535	0.496	0.517	0.605
11	0.436	0.285	x ${}_{f1}$ = 0.633	y ${}_{f1}$ = 0.469	0.517	0.613
			x ${}_{f2}$ = 0.518	y ${}_{f2}$ = 0.568		

**Table 2 sensors-22-08960-t002:** Normalized values of the x and y coordinates of the centroid due to the size of the slice on the sagittal plane.

Set	Patella		Femur		Tibia	
	x	y	x	y	x	y
1	0.0	0.0	0.526	0.441	0.495	0.603
2	0.0	0.0	0.514	0.424	0.512	0.618
3	0.188	0.324	0.475	0.411	0.494	0.651
4	0.234	0.398	0.468	0.385	0.509	0.69
5	0.238	0.399	0.428	0.381	0.492	0.71
6	0.243	0.393	0.422	0.229	0.468	0.708
7	0.241	0.403	0.419	0.286	0.467	0.702
8	0.245	0.411	0.446	0.318	0.484	0.692
9	0.254	0.417	0.453	0.377	0.477	0.665
10	0.3	0.383	0.477	0.42	0.488	0.645
11	0.0	0.0	0.498	0.421	0.507	0.622

**Table 3 sensors-22-08960-t003:** Normalized area values due to the size of the knee on the transverse plane.

Set	Patella	Femur	Tibia
1	0.0079	0.0775	0.2154
2	0.0137	0.0821	0.2157
3	0.0217	0.0901	0.2087
4	0.0348	0.1094	0.2054
5	0.0494	0.1353	0.1997
6	0.0596	0.2298	0.1675
7	0.0615	0.2343	0.1440
8	0.0598	0.2389	0.1343
9	0.0521	0.2428	0.1291
10	0.0387	0.2263	0.1255
11	0.0103	0.0877	0.1150

**Table 4 sensors-22-08960-t004:** Normalized area values due to the size of the knee on the sagittal plane.

Set	Patella	Femur	Tibia
1	0.0	0.0411	0.0174
2	0.0	0.0753	0.0271
3	0.0011	0.1064	0.0526
4	0.0028	0.1178	0.0786
5	0.0125	0.1114	0.1128
6	0.0196	0.0949	0.1084
7	0.0169	0.1006	0.0884
8	0.0076	0.1416	0.0796
9	0.0032	0.1195	0.0608
10	0.0001	0.0892	0.0345
11	0.0	0.0109	0.0233

**Table 5 sensors-22-08960-t005:** Dice index values for the analyzed CT series, for the FC and FCM methods.

Testing	Patella		Femur		Tibia	
Group	FC	FCM	FC	FCM	FC	FCM
1	91.12	90.63	88.24	87.87	90.09	89.89
2	89.37	87.97	86.22	85.63	88.88	87.46
3	87.87	86.29	85.33	84.78	87.13	86.87
4	86.11	84.51	83.81	83.12	84.42	84.18
5	92.08	90.98	89.68	88.48	90.82	89.52
6	90.06	89.01	87.47	86.89	89.34	88.42
7	87.32	87.21	82.66	82.02	84.19	84.09
8	89.82	88.18	87.76	86.31	88.82	87.44
9	89.01	88.87	88.71	86.69	89.67	89.16
10	90.58	90.42	88.01	87.14	90.09	89.68
11	91.36	89.96	87.24	86.78	90.04	89.26
12	88.21	87.02	87.25	86.5	88.89	88.25
13	89.13	88.03	86.19	85.88	88.01	87.73
14	91.44	90.24	88.54	87.26	89.92	88.54
15	92.51	90.87	88.78	87.88	88.78	87.88
16	87.65	86.45	84.38	83.86	86.53	86.28
17	89.87	88.47	87.78	87.18	88.37	88.02
18	85.45	82.85	83.48	82.89	85.24	84.48
19	90.03	89.01	86.44	85.72	87.01	86.21
20	89.36	87.66	84.82	83.64	85.78	84.63
21	87.16	85.76	85.67	84.87	86.29	85.91
22	89.22	87.72	85.52	85.01	87.75	86.32
23	91.31	90.13	88.45	86.92	88.45	86.92
24	86.91	83.97	85.53	83.89	86.34	84.62
25	84.77	82.24	87.18	86.22	87.18	86.22
26	91.41	90.83	86.82	86.21	88.92	87.66
27	92.17	91.67	87.79	86.27	88.58	87.68
28	91.76	91.33	87.34	86.94	87.34	86.94
29	89.23	88.03	85.32	84.82	87.63	86.33
30	89.88	88.67	86.31	85.68	88.23	87.87
31	91.86	90.33	85.4	84.87	87.46	86.9

**Table 6 sensors-22-08960-t006:** Dice index values for the analyzed MRI series, for the FC and FCM methods.

Testing	Patella		Femur		Tibia	
Group	FC	FCM	FC	FCM	FC	FCM
1	90.17	89.67	92.09	90.92	88.67	87.79
2	88.12	88.01	89.9	88.71	86.17	86.11
3	90.36	89.78	92.89	92.16	89.45	86.67
4	86.34	86.1	87.87	86.67	84.12	83.88
5	90.34	90.11	91.42	91.01	89.78	87.23
6	89.37	88.12	88.58	87.17	85.34	85.11
7	87.21	86.89	89.45	88.78	86.67	85.45
8	90.56	89.17	91.18	90.24	89.89	88.17
9	89.1	88.78	92.83	91.67	90.75	89.23
10	86.45	85.34	88.14	86.89	86.13	85.49
11	87.32	86.33	88.33	88.12	86.34	85.24
12	89.31	87.89	88.78	87.15	87.39	85.98
13	89.34	88.1	85.38	84.67	84.2	83.67
14	86.32	86.03	89.34	88.16	86.56	85.43
15	86.78	84.67	86.78	85.67	84.21	84.12
16	89.99	88.45	91.11	90.89	89.98	87.78
17	87.3	86.49	88.18	87.89	86.35	85.55
18	87.68	86.34	89.55	88.67	87.12	85.89
19	89.41	88.45	90.83	89.13	88.56	86.43
20	89.49	88.32	86.33	86.33	85.08	84.36
21	89.28	89.1	89.77	89.34	86.99	85.67
22	85.1	84.73	86.89	85.67	84.21	83.89
23	89.67	87.67	89.13	88.15	86.67	86.34
24	90.01	89.32	90.37	89.67	88.34	87.11
25	90.13	89.34	91.31	89.67	90.11	89.01
26	86.47	85.67	88.37	87.67	86.08	84.79
27	87.89	87.02	87.56	86.89	85.32	84.03
28	86.89	86.03	85.88	84.45	84.41	83.19
29	90.34	89.58	86.87	85.67	84.54	83.46
30	87.28	87.01	86.98	85.89	84.78	83.21
31	90.67	89.02	90.11	88.79	89.19	85.67
32	87.45	86.43	88.37	87.01	86.43	85.89
33	88.24	87.98	90.78	89.16	89.02	87.12
34	86.15	85.12	88.36	88.33	87.1	86.42
35	87.78	86.49	89.13	88.34	86.67	85.98
36	87.34	86.59	86.75	85.78	84.78	83.29
37	86.89	85.51	86.78	85.01	85.41	83.87
38	83.78	83.19	85.98	84.16	84.22	83.1
39	87.45	86.33	89.17	87.56	86.65	85.32
40	86.67	85.41	87.32	86.67	85.11	85.04
41	87.32	85.89	89.03	88.17	87.03	86.12
42	84.69	84.01	86.83	85.03	84.32	83.36
43	87.23	86.2	89.36	88.01	87.18	86.32
44	87.89	86.49	90.12	88.68	88.23	87.32
45	86.32	85.12	88.56	87.07	85.78	84.57
46	86.69	85.23	89.38	87.67	86.21	85.32
47	85.56	84.87	86.37	85.13	85.43	87.76
48	86.22	84.89	87.57	85.75	86.4	86.33
49	86.01	85.31	89.06	86.67	87.54	86.34
50	87.04	85.39	89.17	87.11	87.43	86.47

**Table 7 sensors-22-08960-t007:** *p*-values calculated for the Wilcoxon test and *t*-test for the following bones: patella, femur, and tibia.

Data/Test	Patella	Femur	Tibia
CT: Wilcoxon	0.0159	0.0111	0.0127
CT: *t*-test	0.0172	0.0248	0.0312
MRI: Wilcoxon	not app.	0.0134	not app.
MRI: *t*-test	0.0024	0.0008	0.0006

**Table 8 sensors-22-08960-t008:** Mean, standard deviation (stddev), and *p*-values calculated for the one-sided *t*-test for the following bones: patella, femur, and tibia.

Data	Patella	Femur	Tibia
CT: mean	−0.656	−0.754	−0.754
CT: stddev	1.548	1.640	1.350
CT: *p*-value	0.017	0.025	0.031
MRI: mean	−0.720	−0.869	−0.664
MRI: stddev	0.998	1.221	0.971
MRI: *p*-value	0.002	0.001	0.001

**Table 9 sensors-22-08960-t009:** Dice index values for the 10 tested MRI series (NYU fastMRI Initiative database) for the FC and FCM methods.

Testing	Patella		Femur		Tibia	
Group	FC	FCM	FC	FCM	FC	FCM
1	84.87	83.98	86.23	86.03	85.62	84.78
2	89.13	88.81	90.29	89.75	88.23	87.81
3	88.33	88.10	90.04	89.34	89.79	89.03
4	82.76	81.81	85.56	84.88	83.72	83.09
5	84.45	84.82	86.12	85.67	85.24	84.85
6	84.88	82.96	85.29	84.39	84.47	84.77
7	89.97	88.34	91.42	91.08	90.91	89.79
8	90.01	89.67	91.92	91.11	90.09	89.67
9	79.67	78.58	85.78	84.76	84.34	83.67
10	87.69	86.18	89.23	88.47	88.83	88.26

## Data Availability

All data supporting the reported results can be provided by the author of the article (all MR and CT series after anonymization).

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
