# Peer review of "Atlas-Based Segmentation in Extraction of Knee Joint Bone Structures from CT and MR"

_sensors, 2022, doi:10.3390/s22228960_

Round 1
Reviewer 1 Report
This paper described a two steps approaches to the extraction of bone structures of the knee joint based on atlas segmentation and FC/FCM. This is a very good attempt in improving the segmentation accuracy. While I still have some questions:
1. The knee joint is relatively easy to identify on the image, while the Dice index results for automatic identification in this paper is just under 0.9, please discusse it in the text.
2. References are too outdated. The advantages and disadvantages of deep learning methods are not compared and presented in the text.
3.The pictures and tables in the text are too cumbersome, please streamline.
Author Response
First of all, the author of this paper would like to thank the reviewer for the time and effort put into making an insightful and valuable review.
- The knee joint is relatively easy to identify on the image, while the Dice index results for automatic identification in this paper is just under 0.9, please discusse it in the text.
In the manuscript in the section Conclusion (line 473) the following text was added:
At this point two important problems related to the segmentation of anatomical structures are worth mentioning. The first one relates to the lack of possibility of precise description of the segmented structures (inaccuracy). And the second one concerns identical results, which are difficult to obtain (data uncertainty). The knee joint is relatively easy to identify on the image. However expert’s making two identical outlines of the same bone structures of the knee joint, especially on slices containing the patella, is a very difficult task~\cite{ref-journal1}. Therefore, the values of the Dice index at the level of 0.9 achieved in this work are not low.
- References are too outdated. The advantages and disadvantages of deep learning methods are not compared and presented in the text.
In the manuscript in the section Introduction (line 124) the following text was added:
In the course of the last years, approaches based on Machine Learning and Deep Learning have gained interest. Currently dominating in the practical application is the Deep Learning (revealing high accuracy and fast computational time compared to state-of-the-art methods), with a model of convolutional neural network (CNN) and its architectures in several applications. With regard to the structures of the knee joint the Deep Learning methods were applied to the following anatomical structures of this joint:
-) bone structures: in the paper~\cite{ref-journal16} a fully automatic model has been proposed to detect and segment knee bones using modified U-net models. The obtained accuracy of bone structures has been equal to 98% and Dice index for patella: 92\%, tibia: 96\% and femur: 97\%. In the paper~\cite{ref-journal20} a 2D convolutional encoder network of a Visual Geometry Group 16 (VGG16) architecture (Dice index for femur: 96\% and for tibia: 95\%) has been used.
-) anterior cruciate ligament: in the paper~\cite{ref-journal17} a customized 14 layers residual network ResNet-14 architecture of CNN has been used, with six different directions by utilizing class balancing and data augmentation. The obtained accuracy has been equal to 92%. In the paper~\cite{ref-journal18} a self-supervised approach has been proposed, with pretext and downstream tasks using class balancing through oversampling (accuracy has been equal to 90.6\%). In the paper~\cite{ref-journal19} the AlexNet architecture of CNN to extract features of knee MRNet with transfer learning ImageNet has been presented (accuracy has been equal to 93.7\%). In the paper~\cite{ref-journal21} the knee mask has been generated on the original MRI images to apply a semantic segmentation technique with CNN architecture U-Net (accuracy has been equal to 98\% and Dice index: 99\%).
-) cartilage of the knee: in the paper~\cite{ref-proceeding7} 2D features of CNN have been used for each voxel of three planes (accuracy: 99.9\% and Dice index: 82\%). In the paper~\cite{ref-journal20} a 2D convolutional encoder network of a Visual Geometry Group 16 (VGG16) architecture has been used (Dice index for femoral cartilage: 81\% and for tibial cartilage: 82\%).
Deep Learning methods, apart from their undoubted advantages, also have disadvantages. The right structure of the network has to be chosen. A network that is too small, with no hidden layers, loses its ability to solve problems and even long training is not able to help. Too many hidden layers lead to a simplification of the network performance. Even minor errors in labeling training data, which can occur quite frequently due to human error, can ruin the accuracy of the neural network. In addition, huge amounts of data are needed for training. The training process itself is computationally very expensive, requires a large amount of memory and computing resources, and it is not easy to transfer it to other problems. As a result, the algorithm takes longer to train and more memory is required to work with the data.
3.The pictures and tables in the text are too cumbersome, please streamline.
Figures in the text have been enlarged. The data contained in the tables (Tab. 5 and Tab.6), in order to increase their readability, have been presented on the box-and-whisker and on the BlandAltman plot charts, appropriately (Tab.5 -> Fig.7, 8 and Tab.6 -> Fig.9, 10).
Are the conclusions supported by the results?
In the manuscript in the section Conclusion (line 481) the following text was added:
Based on the obtained results, it can be concluded that the fuzzy FC and FCM methods can be used in the process of segmentation of the bone structures of the knee joint. Atlas based segmentation combined with FC method applied to bone structures of the knee joint gave slightly better (about 1% of Dice index) results compared to the FCM method. The results obtained using both methods are acceptable and satisfactory in medical applications.

Reviewer 2 Report
This manuscript uses atlas-based segmentation using the average feature vector from the automated image-matching method to produce bone structures comprising the femoral head, tibia, and patella. The steps in feature extraction comprise medical image matching using fuzzy image concepts and similarity measures; feature vector extraction at each scan based on training ground truth; scan normalization, and division into 11 sets; averaging features. Fuzzy segmentation methods (FC, FCM) use the average feature vector, centroids (seed points), and surface area (dealing with over-segmentation) for segmenting knee regions. This work achieves good performance on 107 clinical T1-weighted MRI studies of the knee joint and 61 clinical CT studies of the lower limb.
However, this manuscript meets a few issues that need to be addressed as follows:
[1] This work is a continuation of previous studies in [16, 17]. The author should have a comparison of experimental results between the current method and the previous methods on the current dataset.
[2] Figure 3 shows the effectiveness of the method by the number of feature sets by dice score metric and time required (Line 208). What is the time effectiveness?
[3] Many deep-learning methods are used for knee bone segmentation on SKI10, OAI, and ZAB datasets. What is the effectiveness of deep learning methods and the method in this work for the dataset in this work, the other datasets, such as SKI10, OAI, and ZAB?
[4] The maximum year of reference papers is 2018.
Author Response
First of all, the author of this paper would like to thank the reviewer for the time and effort put into making an insightful and valuable review.
[1] This work is a continuation of previous studies in [16, 17]. The author should have a comparison of experimental results between the current method and the previous methods on the current dataset.
In the first mentioned work the main aim was to find the feature vector of the cruciate ligaments. This feature vector had to clearly define the ligaments structure and make their diagnosis easier. The feature vector finding was based on the successive steps in extraction process of both anterior and posterior cruciate ligaments. In the first stage a region of interest including cruciate ligaments (CL) was outlined. The automatic method of location of the CL on the T1-weighted MRI knee images was based on fuzzy C-means (FCM) algorithm with median modification. The next step of that process was the extraction of the cruciate ligament structure using the fuzzy connectedness approach. In the last stage the feature vector was built. The research has been tested on 68 clinical T1-weighted MRI studies.
In the second mentioned work the main aim was to present an automated image matching methodology being used in the field of medicine for inter- and intraobjectional image matching. This paper had shown a different approach avoiding the standard procedures associated with performing four main steps of the registration process: feature detection, feature matching, mapping function design and image transformation with resampling, and replacing them with the fuzzy image concept combined with the use of similarity measures. This methodology has been tested on clinical T1- and T2-weighted MRI slices of the knee joint in coronal and sagittal plane.
Also, the article by Zak, W., Zarychta, P.: Verification of selected segmentation methods in relation to the structures of the knee joint, ITiB 2022, is a continuation of the research. On the basis of the said paper the Region Growing method has been excluded from further research, because the segmentation results were at a much lower level compared to the fuzzy methods described in this paper. And the methodology has been tested on 15 clinical T1-weighted MRI studies of the knee joint.
So, it is difficult to make a comparison between the methods described, because they are dedicated to different anatomical structures of the knee joint, and some require a region of interest, while others cover only a part of the current version of the method.
[2] Figure 3 shows the effectiveness of the method by the number of feature sets by dice score metric and time required (Line 208). What is the time effectiveness?
The division into 11 sets and the determination of the averaged features for the 50 series of T1-weighted MRI of the knee joint takes about 2 hours, and the same process for 30 CT studies of the lower limb takes about 1 hour 50 min. And then the whole process of the femur or tibia extraction for one T1-weighted MRI series of the knee joint takes approx. 34-39 seconds (depending on the number of slices in the series), and for the patella - approx. 32 seconds. All calculations have been done in MatLab R2017a and in computer: Intel(R) Core(TM) i7-2630QM CPU @ 2.00GHz; 4GB RAM.
[3] Many deep-learning methods are used for knee bone segmentation on SKI10, OAI, and ZAB datasets. What is the effectiveness of deep learning methods and the method in this work for the dataset in this work, the other datasets, such as SKI10, OAI, and ZAB?
At this point, none of the deep learning methods have been implemented and tested on an existing data set in relation to the bone structures of the knee joint. A few years ago in the paper: P.Zarychta, P.Badura, E.Pietka „Comparative analysis of selected classifiers in posterior cruciate ligaments computer aided diagnosis”, Bull. Pol. Ac.: Tech. 65(1) 2017, pp. 63-70 a comparative analysis of several classifiers with regard to the posterior cruciate ligaments was performed. In the abstract, it is written as follows: "At the classification stage we employ five different soft computing classifiers to evaluate the feature vector suitability for the computerized ligament diagnosis. Among the classifiers we introduce and specify the particle swarm optimization based Sugeno-type fuzzy inference system and compare its performance to other established classification systems (the artificial neural network multilayer perceptron (MLP), the support vector machine (SVM), the adaptive neuro-fuzzy inference system (ANFIS), the particle swarm optimized fuzzy inference system (PSO-FIS), and the particle swarm optimized ANFIS (PSO-ANFIS). The classification accuracy metrics: sensitivity, specificity, and Dice index all exceed 90% for each classifier under consideration, indicating high level of the proposed feature vector relevance in the computer aided ligaments diagnosis."
An attempt was made to download the SKI10 dataset from the website: https://ski10.grand-challenge.org/Download/, but it proved not possible ("This challenge is closed. You can no longer submit results. You can no longer download the data."). In order to download data from the OAI-ZAB database, you have to register and agree to the following terms and conditions: "No guarantees or warranties for correctness of our data. No commercial use of the provided data. Any distribution of the data, passing on of the data, or sublicensing are not permitted". I am not sure about the legal side of this solution. Therefore, I have to consult these conditions with a lawyer from the Silesian University of Technology.
[4] The maximum year of reference papers is 2018.
The oldest references have been removed or replaced with new ones.

Round 2
Reviewer 1 Report
Thank you for the revisions to the article and for your excellent work.
Author Response
The author of this paper would like to thank the reviewer once again for the valuable and positive review of this manuscript.
Reviewer 2 Report
For issues [1]. [2]. [4], the author explained and revised the work. However, in the issue [3], the author needs to use some deep learning methods in MONAI to make a comparison to this work using the dataset in this work.
Author Response
First of all, the author of this paper, once again, would like to thank the reviewer for the time and effort put into this review.
Retrieving data from the datasets suggested by the reviewer was unsuccessful. The effectiveness of the described method has been verified on 10 randomly selected MRI T1-weighted series downloaded from the website: https://fastmri.med.nyu.edu/.
In the manuscript in the section Results (line 384) the following text was added:
Data (ten MRI T1-weighted series in the sagittal plane) used to verify the results of this study were obtained from the NYU fastMRI Initiative database (fastmri.med.nyu.edu)~\cite{ref-journal22}. As starting points in both fuzzy methods (FC and FCM) were used those determined in this study (Tab.~\ref{tab2}). The obtained values of Dice index for the analyzed MRI series are presented in Tab.~\ref{tab6MRI_FC_FCM_test}. It can be noted that atlas based segmentation combined with the same fuzzy methods (FC and FCM), but applied to bone structures for the fastMRI Dataset (10 T1-weighted MRI studies of the knee joint in the sagittal plane), gave the following Dice index results (average value for the testing group): 86.18 \% (FC) and 85.33 \% (FCM) for patella, 88.19 \% (FC) and 87.55 \% (FCM) for femur and finally 87.12 \% (FC) and 86.57 \% (FCM) for tibia, respectively for FC and FCM methods. These values do not differ significantly from those obtained in the study, therefore the results can be considered as convergent and reliable.